# Wear Analysis of 3D-Printed Spur and Herringbone Gears Used in Automated Retail Kiosks Based on Computer Vision and Statistical Methods

**DOI:** 10.3390/ma16165554

**Published:** 2023-08-10

**Authors:** Jakub Bryła, Adam Martowicz, Maciej Petko, Konrad Gac, Konrad Kobus, Artur Kowalski

**Affiliations:** 1Department of Robotics and Mechatronics, AGH University of Krakow, Al. Mickiewicza 30, 30-059 Krakow, Poland; jakbryla@agh.edu.pl (J.B.); petko@agh.edu.pl (M.P.); kgac@agh.edu.pl (K.G.); 2Independent Researcher, 30-128 Krakow, Poland; kobus.konrad@gmail.com; 3Delfin Sp. z o. o. Sp. k., Ul. Oświęcimska 52, 32-651 Nowa Wieś, Poland; artur.kowalski@delfin.pl

**Keywords:** fused deposition modeling (FDM), fused filament fabrication (FFF), material extrusion (MEX), 3D printing, additive manufacturing, epoxy resin encase, computer vision, statistical methods, gears, wear, automated retail kiosk

## Abstract

This paper focuses on a wear evaluation conducted for prototype spur and herringbone gears made from PET-G filament using additive manufacturing. The main objective of this study is to verify if 3D-printed gears can be considered a reliable choice for long-term exploitation in selected mechanical systems, specifically automated retail kiosks. For this reason, two methods were applied, utilizing: (1) vision-based inspection of the gears’ cross-sectional geometry and (2) the statistical characterization of the selected kinematic parameters and torques generated by drives. The former method involves destructive testing and allows for identification of the gears’ operation-induced geometric shape evolution, whereas the latter method focuses on searching for nondestructive kinematic and torque-based indicators, which allow tracking of the wear. The novel contribution presented in this paper is the conceptual and experimental application of the identification of the changes of 3D-printed parts’ geometric properties resulting from wear. The inspected exploited and non-exploited 3D-printed parts underwent encasing in resin and a curing process, followed by cutting in a specific plane to reveal the desired shapes, before finally being subjected to a vision-based geometric characterization. The authors have experimentally demonstrated, in real industrial conditions, on batch production parts, the usefulness of the presented destructive testing technique providing valid indices for wear identification.

## 1. Introduction

Automated retail kiosks (automated retail vending machines) have become increasingly common, particularly in the USA and Japan where their number per customer is as high as 1 to 23. These machines have transformed into fully automated stores with a product range comparable to small convenience shops. They provide everyday items and offer advantages such as safety, hygiene, and continuous accessibility. Hence, they may be found in various high-traffic areas, e.g., shopping centers, airports, and other transit hubs. The shopping process consists of selecting products with an interactive panel or screen, paying for them, and finally, picking them up using the carts, conveyors, or manipulators. However, it is worth noticing that the application of conveyor belts for transportation of the purchased items maximizes space in the machine and increases the amount of goods that may be stored. Therefore, the issue of exploitation of the polymeric gear components installed in the above-mentioned technical solution is addressed by the authors of the present study.

Additive manufacturing (AM) is gaining popularity in the electromechanical industry due to significant advancements regarding both technical and material aspects. Even more importantly, AM provides the ability to produce nontypical products of complex shapes and structures made from various types of materials at acceptable cost and in a shorter time frame when compared to conventional methods [1,2]. However, AM has some drawbacks, such as limitations for low-volume production and a lack of systematic, and possibly standardized, data on the final properties of the product, e.g., strength, durability, and thermal characteristics. To address these issues, in the case of 3D-printed gears, some authors suggest analyzing the data obtained from the manufactured items rather than making use of the standard specimens [3]. The main disadvantage of AM is the longer production time necessary for a single item, making it unsuitable for mass production. Despite this, recent studies have shown promise for AM in this area [4]. Additionally, the total manufacturing costs, including the costs of AM implementation, are at an acceptably low level; therefore, this technology may be considered a profitable option for manufacturing products.

Specifically, fused deposition modeling (FDM) may be seen as a very attractive and mature fabrication solution. Among other AM methods, FDM is a favorable compromise between the quality of the 3D-printed parts, manufacturing speed, and cost, as well as availability of various types of 3D printers and raw materials (filaments). FDM makes use of an extrusion process performed by a heated filament that gradually creates a target part, layer by layer. Other AM methods, in general, require more expensive and specialized hardware equipment, or a specific form of raw material. In the case of vat photopolymerization, an ultraviolet (UV) light provided by a laser sequentially cures a photopolymer resin which, after hardening, builds a 3D object. UV light is also used in material jetting for hardening a jetted (dropped) material on a build platform. Binder jetting, in turn, creates parts with a liquid binder adhesive provided on top of a powder layer, whereas powder-bed fusion uses an electron beam or laser to melt and fuse material powder. Similarly, directed energy deposition makes use of an electron beam, laser, or plasma arc for melting and fusing material provided in the form of powder or wire. Finally, sheet lamination can be used for sequential bonding of materials’ sheets or ribbons, thus creating a 3D object.

Below, the estimated costs of FDM implementation are briefly discussed. The purchase cost for a FDM 3D printer that enables commencement of production in an industrial environment may be as low as several thousand euros. The cost of a filament can be assumed to be several tens of euros per kilogram of its mass. It should be noted that an alternative technological solution, based on the use of an injection mold, may be initiated spending at least EUR 10,000. Moreover, classical machining processes may be implemented, starting with costs of hundreds or even thousands of euros per single item. Eventually, according to the authors’ crude calculations, with reference to their industrial experience, the estimated production cost for a centimeter-sized 3D-printed part, considering the total volume of 1000 items, may be lower than 10 euro.

Various types of industries, including textile and food production, processing, packaging, and the biomedical industry, often utilize plastic parts because of their cost effectiveness, lightweight construction, resistance to corrosion, and low noise during operation [4]. These components can be used to make cutters, while in electric motors, they are used for covers and gears [5]. Three-dimensional printing provides an easy and efficient means of manufacturing drivetrain system metallic elements counterparts. However, it is worth mentioning that studies on gear trains are typically carried out for metal alloy manufacturing methods. Moreover, even if it is being conducted, the research on gear trains made from polymers, primarily assembled with spur gears, mainly focuses on conventional methods and materials. The researchers discuss modeling-related issues for gear trains, such as polymer component meshing stiffness, the use of numerical and analytical approaches [6,7], and recommendations for polymer gears’ designing processes in order to ensure the desired improvement of the components’ durability and performance [8]. Investigations on the strength of the operationally loaded spur gears made from polymeric materials and the results of studies on their tribological properties are also reported in the literature for cases both with and without lubrication [8,9,10,11,12,13,14].

Polymer gears are usually manufactured by polymer/composite polymer injection molding or machining. Consequently, there is a lack of information on the wear resistance of components made using AM techniques. The exception is, however, the evaluation of their geometry accuracy [15,16,17,18,19]. Many studies relate to the results of strength analyses for 3D-printed specimens and focus on selecting suitable technological process parameters and improving geometric characteristics to achieve increased strength and the desired shape accuracy [20,21,22,23,24,25].

Understanding the wear resistance and durability of the materials used in AM is crucial for avoiding damage from both high temperatures and loads. Several researchers have explored the strength of polymer gears manufactured using 3D printers through both theoretical and experimental means under various operating conditions [26,27,28,29,30]. For instance, Zhang et al. used Nylon, Nylon 618, Nylon 645, and alloy 910 filaments to manufacture spur gears and reported that Nylon 618 was the most durable of the materials tested, outperforming Nylon 66 (PA 66, where PA denotes polyamide) manufactured using plastic injections at low-to-medium torque values [31]. Dimić et al. tested the wear of spur gears made from acrylonitrile butadiene styrene (ABS) and polylactic acid (PLA) materials using AM techniques at different filament extrusion speeds and found PLA to be more durable when compared to ABS [32]. A comprehensive study on the overall quality, wear, and selected production aspects of gears made from ABS, PLA, PA, and PA carbon fiber composite materials was presented in [33]. In the reported tests, the FDM method was employed, which is also known as material extrusion (MEX) [34] and fused filament fabrication (FFF), through which the causes of damage to the tested elements were discovered, i.e., root fracturing of the ABS and PA components and low melting temperature in the case of PLA gears. Moreover, Harsha et al. inspected 3D-printed gears made from ABS, PLA, and nylon at various rotational speeds, and found that the nylon components were the most durable, whereas gears manufactured from PLA were damaged at 1100 rpm [35]. The friction coefficients measured at various temperatures for operating PLA gears were studied by Feteke, who confirmed significant wear reduction (by 35%), achievable by an adequate temperature-related improvement of the mentioned coefficient. As found, as high as a 40% progression of wear may result from a 5 °C temperature increase [36]. Lastly, Tunalioglu et al. conducted wear tests on PLA gears undergoing operational conditions [37].

Summing up and drawing more general remarks on wear, it is worth pointing out several important perspectives for reduction via modification of design and fabrication processes. First, Farooq et al. investigated the influence of the selected characteristics of utilized manufacturing processes on the operation of fabricated components and their gradual loss of initial properties, i.e., aging [38]. Electric discharge machining was considered of particular concern when studying the machining tool and process-induced improvements of tribological properties that, in turn, result in a reduction of the intensity of wear mechanisms. Other recent work [39] has focused on material selection for kinematic pairs that establish a long-term and hard-on-hard type of interaction. As proven, an improper choice of materials for constructing the inspected interacting parts may disadvantageously accelerate their wear processes. Finally, results from a study on the influence of geometric characteristics of the operation and wear of fabricated parts are reported in [40]. The authors of the referenced work, making use of the identified contact pressure, attempted to reduce the wear, performing several shape modifications, including adaptation of both the linear and angular dimensions.

This paper presents the results of a wear test conducted on spur and herringbone gears manufactured via 3D printing using the MEX technique, which were then used in a coupling mechanism in an automated retail kiosk. The reported study began with measuring the basic geometrical parameters of the gears before and after the test, using visual methods to detect and track damage progression. Additionally, the mobile cart’s position and velocity plots, as well as the torque plots generated by the system’s motors, were analyzed to identify impending gear failure. The objective of the present work was to determine whether 3D-printed gears can be a reliable replacement for conventional metal gears and composite polymer injection-molded gears. The automated retail kiosks designed by Delfin Sp. z o. o. Sp. k. were used to test this hypothesis. This study also investigated the use of basic statistical methods to analyze kinematic parameters and torques, obtained from internal sensors used to monitor the implemented drives, to predict the wear level of plastic gear trains. Finally, this paper aimed to contribute to the limited knowledge on wear in gears made using the MEX technique.

The contribution presented in this paper, which is considered to be novel, is the complete approach proposed for identification of the changes of the geometric properties of 3D-printed parts resulting from wear. Specifically, the analysis integrated several steps that allowed for the above-mentioned geometry identification. First, a 3D printing process was used to create the investigated parts following their designed shapes. Next, for the selected subset of manufactured parts, their exploitation under the standard operational conditions was experimentally investigated while preserving a collection of non-operated items for reference measurements. Then, for each of the examined components, they were completely encased in resin and cured. Following this, 3D-printed parts were cut in a specific plane to reveal the desired cross-sectional shapes. Finally, vision-based geometric characterization for both exploited and non-exploited components was performed to conclude about their wear. The authors do not know any similar measurement and wear quantification methods reported in the literature. Complementarily, the authors are aware of the fact that embedding 3D-printed parts in resin itself is a known technique and is willingly used to enhance their strength properties, as reported in [41], but this technique has not previously used to improve their geometric characterization. What seems even more interesting is that the authors of the current work demonstrated that the presented destructive testing technique provides valid indices for wear progression. This is possible even though simultaneous investigation of various operational conditions and a gradual increase in working periods performed for the same printed item are not feasible. In fact, once being examined, a given specimen cannot continue its exploitation or be subjected to the measurement process again. As shown in the present paper, advantageously, the destructive nature of the proposed method does not prevent us from formulating valid conclusions regarding wear. The raised capability should be considered of great practical importance in the field of material strength investigation for 3D-printed parts used in industrial applications.

The industrial use of elements made with AM, as well as the search for new methods dedicated to assessing the change in their technical condition (health) resulting from operational conditions and wear, are currently issues of high practical importance. Nowadays, the high availability of relatively low-priced AM machines offers an attractive alternative for the production of elements at both the prototype stage and in the final applications for the construction of machines and mechanisms. Any improvement regarding the testing procedures elaborated for mass-produced 3D-printed devices is highly desirable, as in the case identified for the current research. It is worth noting that in the past, 3D printing technology was mainly used to produce housings and other components that do not play an important role in the operation of machines. In contrast, the rapidly evolving field of AM allows for the creation of more advanced and durable mechanical components that are critical in many types of exploited machineries. Among other applications, the use of 3D printing to produce gears is particularly interesting because these components are typically subjected to heavy loads and require high durability during their standard long-term operation. This issue is a challenge in terms of the proper selection of both materials (or raw materials) and machineries used for fabrication.

## 2. Materials and Methods

In this study, a wear analysis of 3D-printed gears working in a commercial automatic retail kiosk was performed. Two types of gears were tested: Straight-cut (spur) and V-shaped teeth (herringbone). The spur gears can be characterized as follows: Number of teeth, *n* = 25; outer circle diameter, *D* = 54.00 mm; inner circle diameter, *d* = 20.00 mm; modulus, *m* = 2.00 mm; circular pitch, *p* = 6.283; addendum, *ha* = 2.00 mm; dedendum, *hf* = 2.40 mm; total tooth depth, *ht* = 4.40 mm (sum of the addendum and dedendum) (Figure 1). Similarly, the herringbone gears can be characterized as follows: Number of teeth, *n* = 25; outer circle diameter, *D* = 54 mm; inner circle diameter, *d* = 20.00 mm; helix angle, *ψ* = 9.13°; modulus, *m* = 2.00 mm; normal circular pitch, *pn* = 6.283; addendum, *ha* = 2.00 mm; dedendum, *hf* = 2.40 mm; tooth whole depth, *ht* = 4.40 mm. A filament of 1.75 mm in diameter, made from polyethylene terephthalate glycol (PET-G)-modified, was selected as the material for manufacturing the gears. The main requirements for the selection of a 3D printer were a sufficient workspace size, closed working chamber for stable printing conditions, low cost of single-component fabrication, and high availability. From the 3D printer models fulfilling these requirements, the ATMAT Signal Pro 3D printer was chosen. This printer exhibits a large workspace with dimensions of 300 × 300 × 500 mm^3^. This capability was considered important for the investigations on the properties of the prototyped components and made it possible to simultaneously print items of various sizes and shapes. The resulting increased effectiveness of the 3D printer also allowed for fast fabrication of many components at a time and finally decreased the manufacturing time period and costs. Finally, the fast production track allowed to safely keep the reserve stock continuously available for installation in the operating automated retail kiosks.

Referring to the 3D printing characterization found in [42,43], the most important process parameters considered in the current study are as follows: first, it should be noted that the slicer CURA 4.4.0, executed for its standard MEX technology settings regarding extrusion nozzle diameter, number of perimeter layers, and layer thickness, was used to generate G-code. Specifically, the gears were printed using a 0.4 mm nozzle diameter, with a 0.3 mm layer thickness, three outline perimeters, and a 100% infill density with the application of a linear pattern. During filament extrusion, swapping of the arrangement of the fibers in successive layers by ±5 deg. was applied. The filament was melted and extruded at 245 °C, while the build platform (heatbed) was heated up to 65 °C and its temperature sustained throughout the whole production process. The estimated mass of a printed sample equaled 35 g, while the approximate printing time period was 3 h 20 min.

As shown in Figure 2, the gears (14 and 24) were mounted on a mobile cart (3), making the link between the shaft of the AC servomotor (23) on one side and a conveyor belt’s passive shaft (12) on the other. The quality control for a tested gear was carried out directly on a sleeve of the shaft located on the mobile cart that, when coupled with the conveyor belt gear, drives the entire system. The design of the gear’s socket ensured a fixed position while fitting. The gear was locked in place on the sleeve using a machine tapper, i.e., a mounting system of conical male–female socket form, that generated a high friction force when tightened. Consequently, the torque from the drive gear was transferred to the passive belt gear. During operation, if the gear slipped from the sleeve, confirming loose fitting, the printout geometry was determined to be incorrect, and the gear was discarded. Two servomotors (Beckhoff AM8000 series) were used to move the cart’s *X* and *Z* axes, providing the necessary coupling with a conveyor belt (Figure 2). Once the gears were coupled and a gear train was established, the servomotor was responsible for activating the conveyor belt and transporting a product chosen by the buyer. A Beckhoff EL7200 series servo system was used for the *X*-axis and AX8000 series for the *Z*-axis to measure position, velocity, and torque. The machine was located in an industrial hall to prevent significant temperature changes (operation at room temperature), depending on the time of day and night.

To assess potential damage (wear) of the gears, a test of 25,000 cycles of continuous operation, equivalent to one-year-long standard exploitation, was proposed. The measurements were conducted in 44 sessions per gear, separately consisting of 18 consecutive cycles, involving:Movement of the mobile cart from an idle position to a selected conveyor belt;Coupling of the mobile cart to the selected conveyor belt (gear coupling);Continuous work of the gears for a defined period of time (gear train established);Uncoupling of the mobile cart from the conveyor belt (gear uncoupling);Movement of the mobile cart to the idle position.

The cycles were registered at predetermined time intervals. The mobile cart continuously interacted with the same conveyor belt.

### 2.1. Vision-Based Inspection

To determine the wear level of the exploited gears, the authors proposed the measurement of their basic geometrical parameters using visual methods, and later comparison with those of their not yet exploited counterparts. Images for use in further analysis were acquired using a Keyence VHX 7000 digital microscope stand, as shown in Figure 3.

Having considered the complicated geometry of the used herringbone gears (shown in Figure 4), the authors proposed an original method of encasing all of the gears in resin and then preparing cross-sectional cuts perpendicular to their axis via a machining procedure to be able to eventually extract their shapes during vision analysis.

For encasing the elements fabricated via the MEX technique, epoxy resin POLYCORE/Crystalline 940 was applied. Hence, the authors made use of its advantageous properties as a Duroplast material (thermoset). This plastic material is densely cross-linked, also exhibiting hard-elastic areas. It is worth mentioning that for this type of resin, softening does not cause any significant weakening of the internal links, as observable in other plastics. Therefore, the used epoxy resin was not exposed to plastic deformation due to the increased temperature generated during the cutting process. The main characteristics of the used resin, provided by the distributor, are as follows: Thermal degradation above 200 °C and heat deflection temperature (HDT) of 94 °C, whereas the HDT for the PET-G remined within the range of 62–68 °C. Summing up, a raising temperature has lower influence on the material response (plastic deformation) for the used epoxy resin than that observed for the PET-G filament. In the experiments, low-temperature machining was used. Additionally, as experimentally confirmed, the resin thoroughly covered the gears and protected their edges against local stresses and their small fragments from breaking off.

The desired effect of retaining the original shapes of the cross-sectional areas for the parts being cut was achieved by a special approach to the machining process. Preparation of all samples for vision-based testing was commissioned from a company specializing in cutting and machining, including plastic materials. The main guidelines formulated by the authors prior to material processing stated that the glass transition temperature for PET-G, which is approximately 85 °C, cannot be exceeded. Finally, the temperature limit of 70 °C for the machining process was not exceeded, as confirmed by the company. Moreover, according to the information received from that company, an appropriate tool was used to process the PET-G material. First, the shape of the tool ensured continuous chip removal. Additionally, the desired formation of flowing or ribbon-shaped chips was guided. Hence, effective heat energy dissipation from the processed sample via chips was achieved during machining. This allowed us to protect the sample from damage due to excessive heating. Second, the machining tool used was made via sintered carbide technology adapted to the processing of plastic materials, including PET-G. Moreover, the machining parameters, including cutting speed (which was within the range 300–500 m/min) and feed speed (0.05–0.5 mm/revolution), were appropriately selected considering the geometry and type of tool used and the type of processed material. Finally, additional air cooling was also provided.

The applied resin provided a background for vision-based identification of the gears’ cross-sections. Therefore, the color of the resin was found to be crucial for further analyses. Specifically, the colors of the resin and gears should contrast with one another in order to ensure the best possible results are obtained by vision methods. To determine the appropriate color for the analysis, four auxiliary gears were consecutively encased in white, yellow, green, and red resins. In each case, the same base epoxy resin was used, with 3% mass contribution of the selected dye. Figure 5 shows the obtained cross-sections of the gears, which—for the preliminary study—were cut by hand and sanded using an angle grinder. Based on the initial studies, the best contrast effect was obtained for the gears encased in the green resin.

Next, 12 gears encased in green epoxy resin were prepared, namely, four sets of three gears each, including exploited and non-exploited items of both spur and herringbone gears. Using a CNC machine, the surfaces of the tested samples were planned, and cross-sections were made ready to be applied for vision-based measurements. An example image taken by the digital microscope stand is shown in Figure 5. All images were registered during a single measurement session.

The gears’ geometrical parameters were estimated using a five-step computer vision algorithm, implemented in Python software ver 3.10 using OpenCV library. In the first step, a scale template was identified in the image by applying a red calibration mask. The scale factor in the unit (px/mm) was determined by counting the maximum number of red pixels in a row, which was performed separately for each image of the gear. The scale template in the images represents a length of 10.00 mm. The factor identified during all of the conducted measurements took values within the range of 214.6–214.9 px/mm. The scale stencil was then removed from the image by creating a mask for the red color and performing conjunction operation (script cv2.bitwise_and) on both images, i.e., the original and the mask. The results of the above-described operation are shown in Figure 6.

In the second step, the color space was converted from RGB to HSV coordinates, and a mask was made for a given range of HSV coordinates to further remove the background corresponding to the green resin. The areas shown as black in the mask were removed from the original image. The operation of removing undesirable areas from the original image was carried out using the mask, similarly to the first step of the analysis. Adequately, the conjunction operation was used. The obtained results are shown in Figure 7.

In the third step, the images were converted to greyscale, and noise reduction and binarization were performed using the morphological closing operation. The consecutively processed images are shown in Figure 8. The current stage allows for removal of bright reflections on the outer contours of the gears.

In the fourth step, a morphological gradient/skeleton operation was performed and the cv2.findContours library function was used to identify the contours of the objects in the image. At the same time, the lengths of the contour perimeters were analyzed, from which the two largest were determined as the gear’s outer (marked in blue) and inner (in green) contours, respectively, as shown in Figure 9. Using the cv2.minEnclosingCircle function, the centers and diameters of the smallest circles describing the two contours, were obtained, which represent the outer (addendum) and inner (base) circles of the gear, respectively. The results obtained after application of the above-mentioned functions are shown in Figure 9.

In the fifth step, two additional diameters of a given gear were determined: Pitch circle diameter (the dimension marked as DE in Figure 1) and root circle diameter (parameter DT). Having identified the outer circle diameter of the element, a circle with a radius reduced by 1 px was drawn in the subsequent iterations. Using conjunction and inversion operations, the mentioned circle was circumferentially divided into two categories of arcs: Those intersecting with the gear (indicating teeth, marked in green) and those not intersecting (indicating grooves, marked in red). Figure 10 refers to a single iteration of the considered loop operation. When the sum of lengths representing intersecting and non-intersecting arcs were equal, the pitch circle diameter was determined. In the case when the total length of the intersecting arcs exceeded 90% of the circle’s circumference, the former parameter was considered as the root circle diameter. The final results are shown in Figure 11.

The measured values of the gears’ geometrical parameters, acquired by means of the proposed computer vision algorithm, are shown in Table 1 and Table 2 in Section 3, followed by a discussion.

### 2.2. Investigation of the Kinematic Parameters and Torque

Due to the industrial specificity of the utilized experimental setup, the available data are limited. The measured quantities are the selected kinematic parameters and torque found for the two servo motors. These outputs describe the cart movement in the *XZ*-plane (as visualized in Figure 2), i.e., perpendicular to the tested spur gear’s rotation axis. The three data channels comprising position, velocity, and torque were recorded for each servo motor. The registered sample rate was 100 Hz.

During continuous operation of the automated retail kiosk, the data sessions were recorded in one-hour intervals, in 18 cycle runs, spanning an average of 1 min of the cart’s operation, while being coupled to the conveyor belt. Each of the gears was tested for 44 h, simulating the yearly period of normal work of the machine. Three spur and three herringbone gears were tested, resulting in six sets of raw data, with 44 sessions each.

The system strictly followed the set drive path and no significant deviation was observed as shown in Figure 12 and Figure 13. Therefore filtering of the relevant data points corresponding to the spur gear coupling and work could rely on the position and velocity data channels.

The correspondence between position and velocity was used to filter out the gear coupling phenomenon, and hence decoupling useful samples from the raw data (Figure 14 and Figure 15 present examples of the registered data). The cart’s idle movements were filtered out by masking data points with a velocity greater than 0.01 mm/s with 0.01 mm/s tolerance. The cart’s set coordinates were masked by limiting the position to a 0.01 mm proximity to the work point at the position (*X* = 43 mm, *Z* = 625 mm).

A debouncing step, using the arithmetic mean with a window size of 10 samples (which corresponds to a time period equal to 100 ms), was used to preprocess the data before passing them on to the filtering step.

The torque for the *X* and *Z* axes was used for further analysis. However, the torque data channels were omitted during the data filtering step, as presented in Figure 15, to preserve most of the information that has the potential to indicate any changes or wear in the gear drive mechanism.

The torques registered using distribution plots of the *X* and *Z* axes of all three types of spur and herringbone gears, for all 44 sessions, are shown in Figure 16 in Section 3.

## 3. Results and Discussion

The measured values of the gears’ geometrical parameters, acquired by means of the proposed computer vision algorithm, are shown in Table 1 and Table 2. The presented experimental results are complemented with their design counterparts, provided for reference.

First, the authors made some important observations regarding deviation between the fabricated samples (characterized with the data presented in Table 1 and Table 2) and the design (declared by the nominal parameters specified in Section 2). Specifically, the experimentally identified smaller inner and outer circle diameters *d* and *D*, compared to their design counterparts while maintaining mutual centricity, suggests the appearance of calibrating errors for the used 3D printer. This possibly relates to the improperly declared ratio between the number of generated motor steps and the resulting linear feed along the *X* and *Y* axes. However, the difference in the tooth depth with respect to the nominal value was less than 0.2 mm, which seems the typical accuracy for MEX 3D printers. In view of the above observations, only a slight improvement regarding dimensional reproducibility may be effectively achieved by applying a calibrating procedure for the stepper drives. In the authors’ opinion, it may by concluded that the identified geometric inaccuracy mostly resulted from the characteristics of the used 3D printing technology, i.e., MEX, including the specificity of the employed 3D printers that operate using open loop trajectory control algorithms. Summing up, the authors would like to emphasize that the main objective of the present work was, however, verification of whether the used MEX technology, considering its specific properties, accessible geometric accuracy, and material, can be successfully applied to fabricate gears. As found, none of the above-mentioned aspects were identified as preventing the desired industrial capability.

When encountering one another, gears make sliding and rolling movements. As a result, wear occurs on the surfaces of the gears. This phenomenon of complicated physics reveals various features in different areas, depending on the contact ratio of the teeth. The wear that is observed on a tooth surface varies depending on both the sliding speed from the dedendum to the addendum and the distribution of the load on the gear pair.

During operation, gears make double tooth contact in regions near the root of the tooth (i.e., where the gear enters the clutch) and the addendum region (where the gear leaves the clutch), depending on the contact ratio. The load is jointly carried by the two teeth and, therefore, the wear is less intense in these regions. The region on the tooth that is closest to the pitch circle is a single tooth contact area, where the load is not sheared and higher wear is expected. However, the geometrical parameters measured for both exploited and unexploited gears varied by less than 1%. Generally, no significant changes in the geometry of the gears were found as a result of the tests emulating their annual operation in the automated retail kiosk.

Considering the range of the inherent uncertainty regarding the values of the geometrical parameters, measured using the proposed visual method, resulting from the algorithm settings, it should be noted that the differences in the parameter values of unexploited and exploited gears were at the level of their estimation error.

At the same time, considering the high quality of the analyzed sample images and the differences in the parameters of both sets of gears, in the range of tenths of a millimeter, the wear of the gears was negligible.

Considering the results of the application of the second method used in the current study, it is crucial to note that changes indicating possible wear of the gears were observed in the torque distribution, mainly in the standard deviation of the *Z*-axis torque. When preparing the experimental data for discussion, the authors also observed that the first herringbone gear tested exhibited a specific noticeable linear regression of the torques in the *Z*-axis, as shown in Figure 16. Still, it was discarded as an edge case attributed to the cart’s coupling setup time for the newly assembled device. For the rest of the data sessions, there were no noticeable changes in the average torques in either the *X*- or *Z*-axis.

Figure 17 and Figure 18 present the torque distribution plots for 44 measurement sessions performed for the three spur and three herringbone gears tested. As already mentioned, the subsequent sessions referenced the analyzed stages of simulated long-term exploitation. For the experimental results presented in Figure 17 and Figure 18, it was found that the standard deviation of the torque increased with the operation time, and noticeable changes in the number of peaks in the torque graph can be seen with increased wear of the spur gears. The annotations for the torque distribution plots are provided in Figure 19, specifying the selected quartiles and percentiles and median and mean values for more convenient data interpretation.

As identified, at the beginning of the work, the gears had smooth unimodal torque distribution, with a singular peak, whereas near the end of the test period, a multimodal torque distribution started to appear. Multiple peaks started to develop, with a noticeable increase in the RMS value for the torque distribution, which are shown in Figure 20 and Figure 21. Complementarily, the torque profiles for the *Z*-axis identified for the two types of gears are presented in Figure 22.

The idle position’s torque should fall to zero in the horizontal direction (along the *X*-axis) and remain constant in the *Z*-axis when the cart is not moving and is only influenced by gravity. However, the variance in the *X*-axis torque in the idle position observed in Figure 20 and Figure 21 does not show a correlation with the torque in the work position. From this, it can be inferred that the mean torque changes between sessions did not have a high enough signal-to-noise ratio to reach a conclusive statement; thus, only the torque profile in the *Z*-axis could be used to assess the wear of the spur gear and was used for further analysis.

The increase in the standard deviation of the *X*-axis torque could be attributed to uneven wear of the spur gear teeth. The drop in the *Z*-axis torque in the final sessions of the test run is a good predictor (symptom) of increased wear and growing beating of the spur gear mechanism. The change in the number of peaks could be attributed to wear of the teeth, with material displaced from the pitch point of the driving spur gear.

## 4. Conclusions

Two methods of assessing the technical condition of 3D-printed gearwheels were proposed, implemented, and tested: A destructive (vision-based) and a non-destructive (based on tracking kinematic parameters and transmitted torque). The authors’ intention was primarily to test these methods using the newly created prototype built with the gears. The vision-based method applied to gearwheels, i.e., the new ones and those exploited equivalently to their annual operation, showed no significant wear, confirming that 3D-printed parts can replace metal ones in automated retail kiosks. The described research indicates that the profile of the torque transmitted by a gearbox seems to be a good candidate for a symptom of the technical condition of 3D-printed gearwheels and allows to track their wear in a nondestructive way, using measurements already taken by the drive controllers, which is a feasible on-site solution.

The authors are aware of the fact that the behavior and parameters of the same gear, especially before and after exploitation, have not yet been studied. Additionally, a limited population of gears was used during the preliminary analysis. The reported research made use of only a few gears, separately with intact and damaged objects, to primarily allow for presentation of the developed technique dedicated to characterization of the technical conditions in an industrial case.

The introduced measurements help to track the actual progress of the wear based on registered data being processed with vision-based methods, as well as acquired kinematic properties and torque. It is, however, worth noting that operation under conditions allowing for changing the prototype’s input may make it difficult to extract (decoupling) the behavior change of that prototype due to the fluctuations of various operational factors that simultaneously contribute to the resultant characteristics of the inspected system. Hence, in the authors’ opinion, further experimental analysis is recommended for the prototype to enable more comprehensive description of both the relationships between the input parameters and their influence on the system.

Conducting research on the technical state of 3D-printed elements performed after a given operational period opens new possibilities in the field of design, manufacturing, and exploitation of precise and durable mechanisms such as gears. On the contrary, the authors’ experience shows that the use of AM technology for the construction of these types of components is still not common, which may result from insufficient systematic information and guides about the operational capabilities of the extensively exploited 3D-printed elements. In general, the results of studies such as those presented in this paper should be considered to significantly and advantageously affect the development of AM applications in various industries, from machinery to medical or automotive devices.

As for the future development of the presented approaches, the authors plan to repeat the measurements and data analysis sequence with temperature changes. Consequently, an extended inference on the tested system’s operational characteristics and wear could be conducted. Moreover, a continuation of the current wear study aided with computational simulations, e.g., as presented in the work, could be considered [44]. Consequently, the authors plan to take advantage of the lower costs of the conducted investigation and faster acquisition of the results compared to experimental testing. The studies [45,46] reported comprehensive computational frameworks that are suitable for modeling the phenomenon of wear and for determining mechanical and strength characteristics. The cited works consecutively addressed the declaration of geometric and material parameters and the elaboration of a finite element model with handled boundary and initial conditions, which were finally complemented with a definition of adequate load cycles referring to the specificity of the studied research case.

## Figures and Tables

**Figure 1 materials-16-05554-f001:**
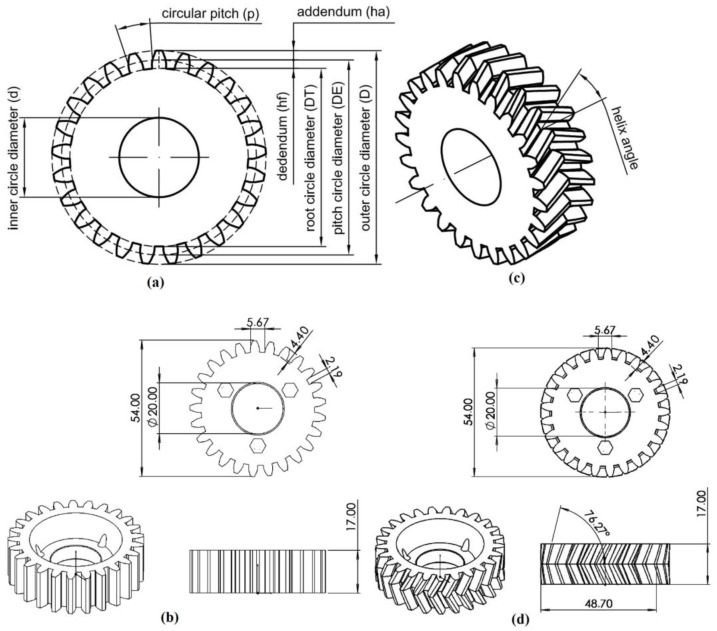
Spur gear: Definitions of the geometrical parameters (**a**) and their values for the designed 3D-prited prototypes—design parameters (**b**); herringbone gear: Definition of the helix angle (**c**) and design parameters (**d**).

**Figure 2 materials-16-05554-f002:**
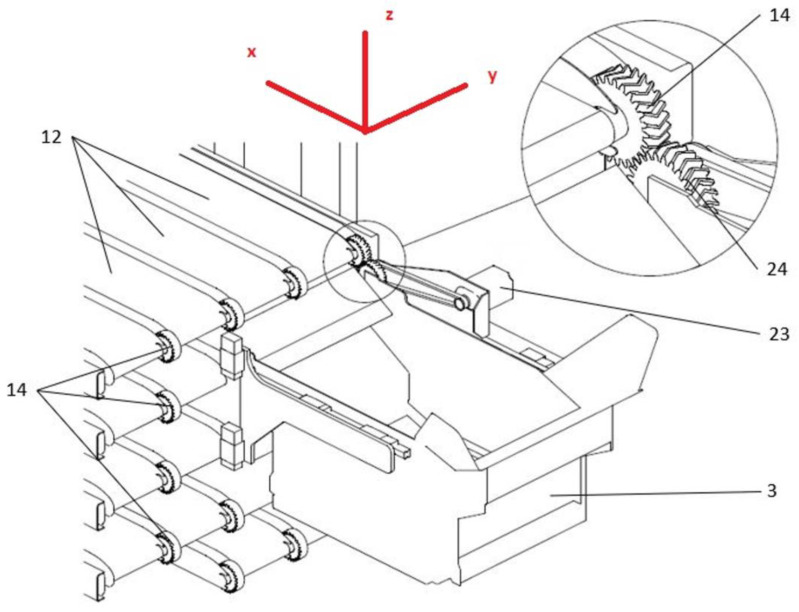
Schematic view of the automated retail kiosk’s inner mechanisms: Mobile cart and conveyor belts.

**Figure 3 materials-16-05554-f003:**
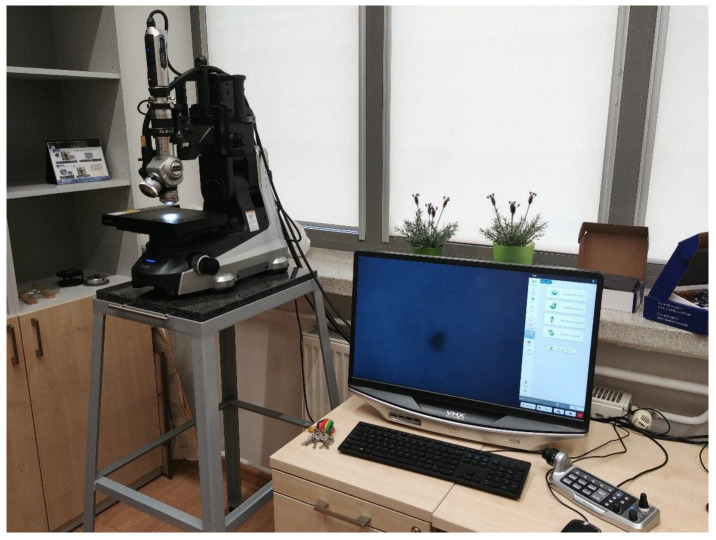
Keyence VHX 7000 digital microscope stand.

**Figure 4 materials-16-05554-f004:**
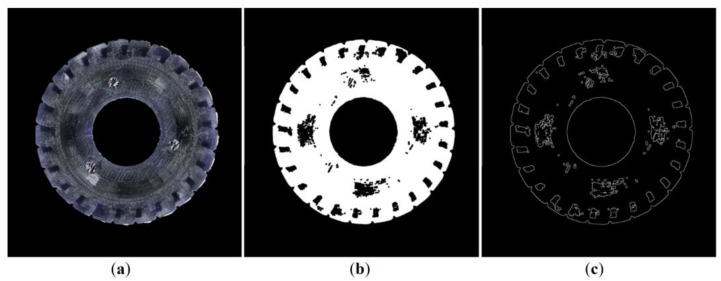
Cross-section image of the herringbone gear (**a**), as well as binarization (**b**) and skeleton gradient (**c**) errors. The gear’s helical teeth prevent it from being correctly processed by the proposed computer vision algorithm.

**Figure 5 materials-16-05554-f005:**
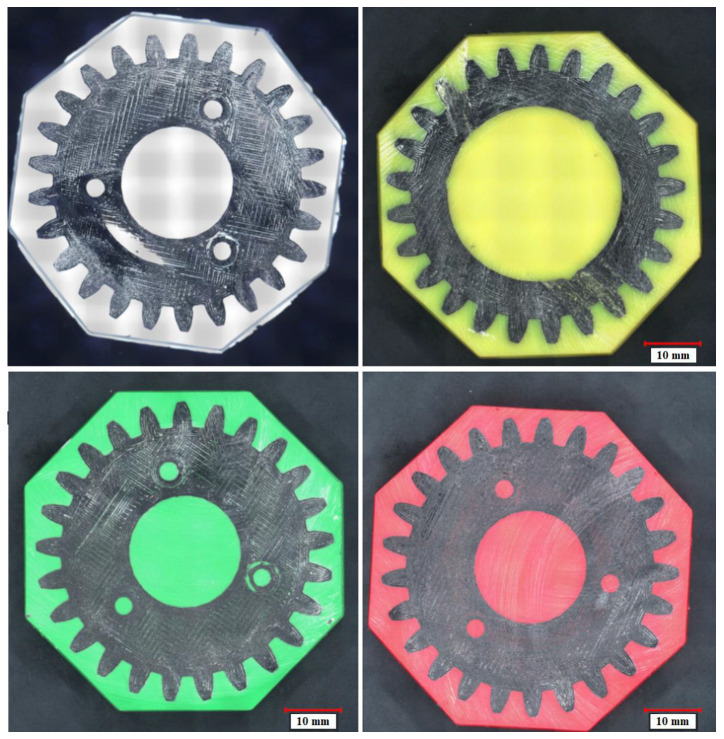
Cross-sections of the auxiliary gears encased in white, yellow, green, and red epoxy resin.

**Figure 6 materials-16-05554-f006:**
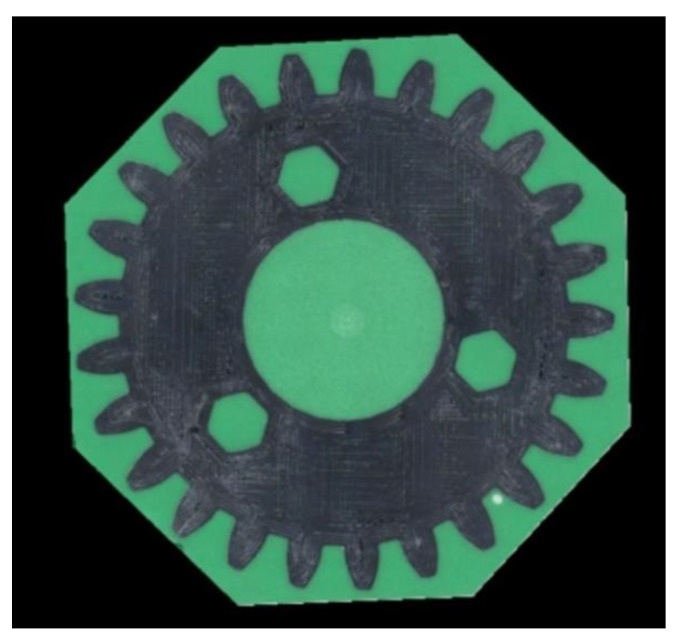
Result of the first step of the algorithm: Spur gear image with removed scale template.

**Figure 7 materials-16-05554-f007:**
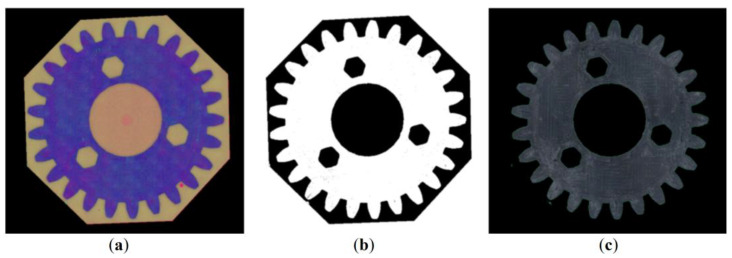
Second step of the algorithm: Color space conversion from RGB to HSV (**a**), removal of the background corresponding to the green resin (**b**), and removal of black areas of the mask on the original image (**c**).

**Figure 8 materials-16-05554-f008:**
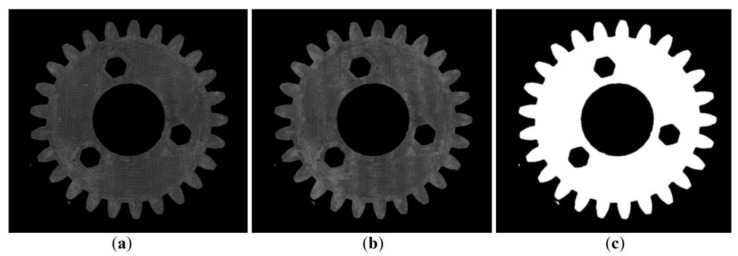
Third step of the algorithm that allows for removal of bright reflections and sequentially covers grayscale conversion (**a**), noise reduction (**b**), and binarization (**c**).

**Figure 9 materials-16-05554-f009:**
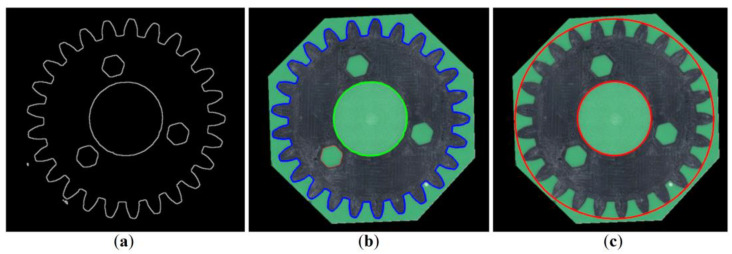
Fourth step of the algorithm: Morphological gradient/skeleton (**a**), identification of the two largest outlines (**b**), and circles describing outer and inner outlines of the gear (**c**).

**Figure 10 materials-16-05554-f010:**
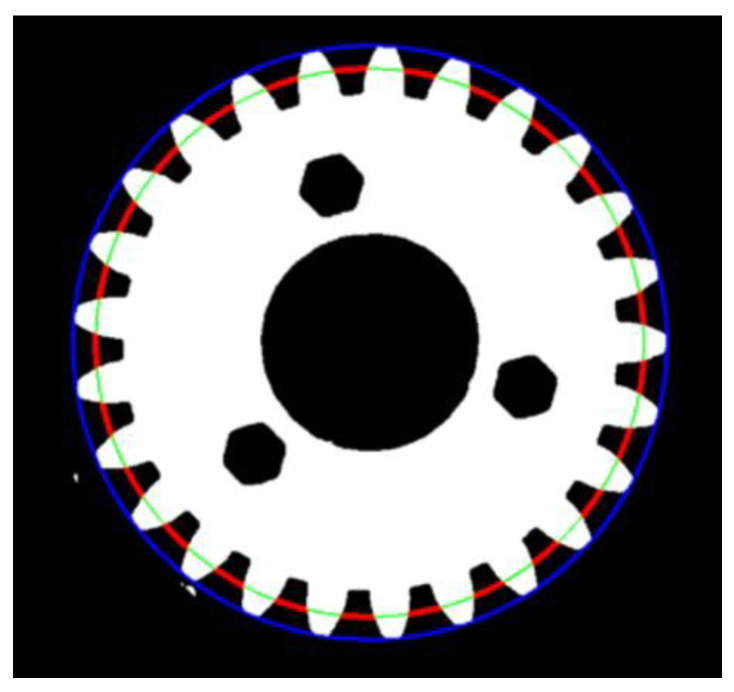
Fifth step of the algorithm: Iterative method determining the pitch circle diameter (when the total length of the green arcs equals the total length of the red arcs); the pitch (green) and outer (blue) circle diameters are drawn on the gear’s image.

**Figure 11 materials-16-05554-f011:**
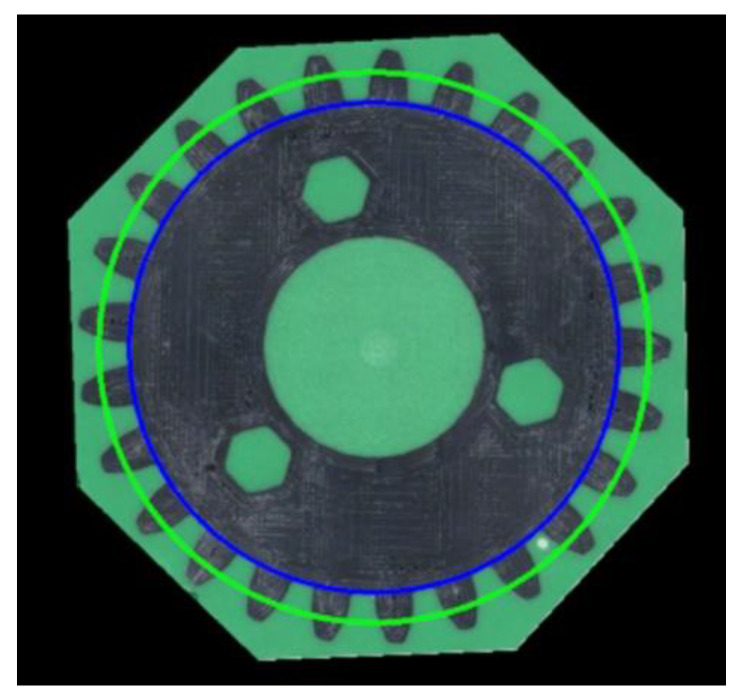
Fifth step of the algorithm: Iterative method determining the root circle diameter (when the total length of the green arcs exceeds 90% of the circle’s circumference); pitch (green) and root (blue) circle diameters are drawn on the gear’s image.

**Figure 12 materials-16-05554-f012:**
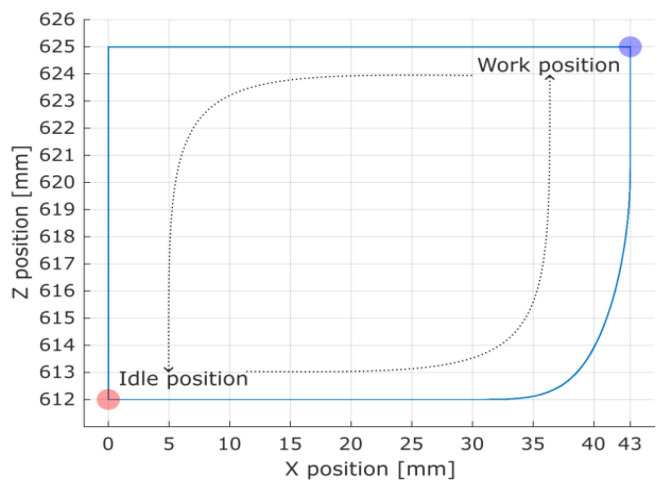
Overlapped plots for all session trajectories in the *XZ*-plane for a single spur gear tested. The cart was moving from the idle point to the work point along the lower path and back along the upper path.

**Figure 13 materials-16-05554-f013:**
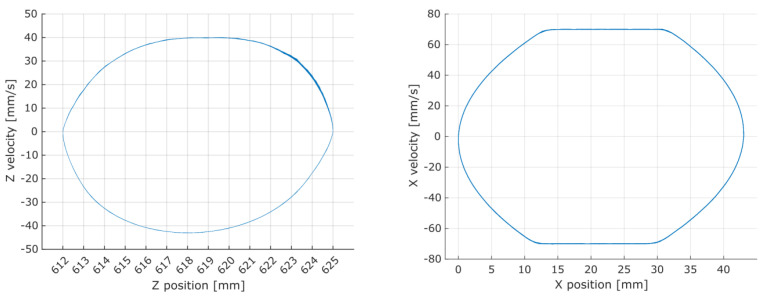
Overlapped plots on the phase space diagrams for the trajectory of the *X* and *Z* axes for a single spur gear tested. The meshing of the spur gear teeth is visible in the *Z* velocity channel near the coordinate *Z* = 625 mm.

**Figure 14 materials-16-05554-f014:**
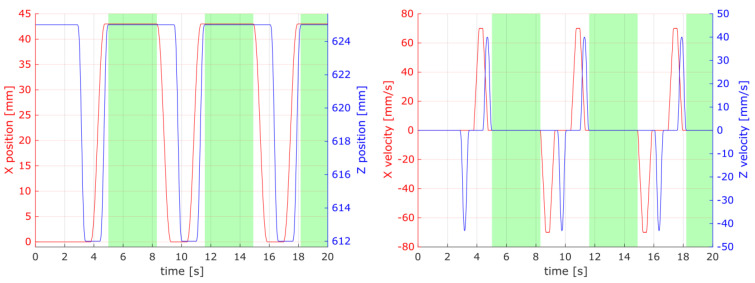
Raw data for the position of the *X* and *Z* axes and the velocity channels. Highlighted in green are the data ranges that passed the filtering steps.

**Figure 15 materials-16-05554-f015:**
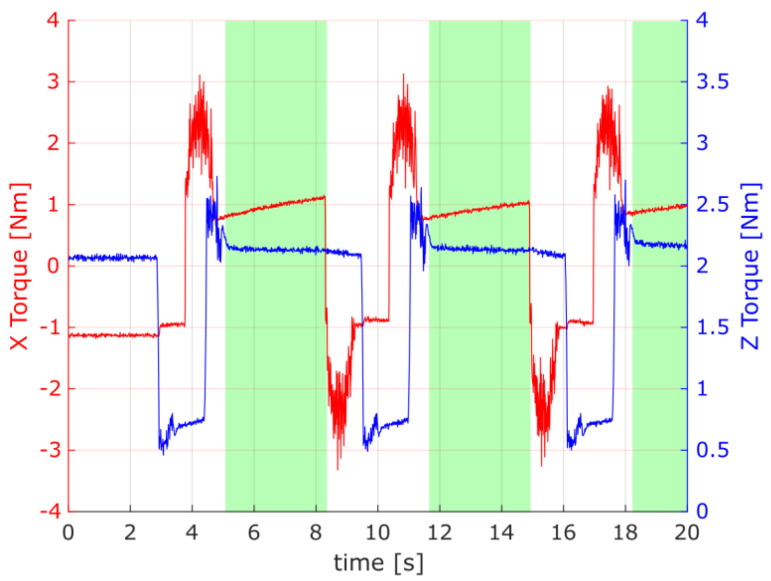
Raw data for the torque channels of the *X* and *Z* axes. Highlighted in green are the data ranges that passed the filtering steps.

**Figure 16 materials-16-05554-f016:**
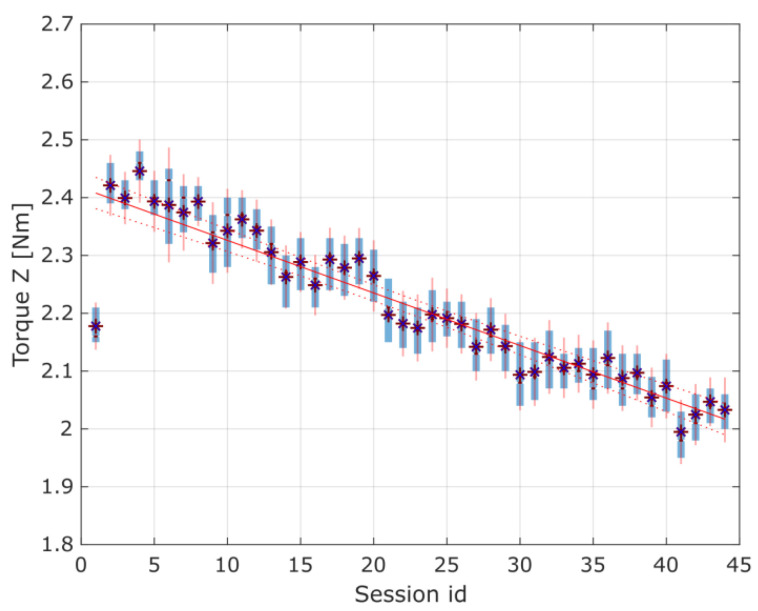
Torque in the *Z*-axis for the first herringbone gear tested with linear regression fitted. A detailed explanation regarding data presentation is provided in Section 3.

**Figure 17 materials-16-05554-f017:**
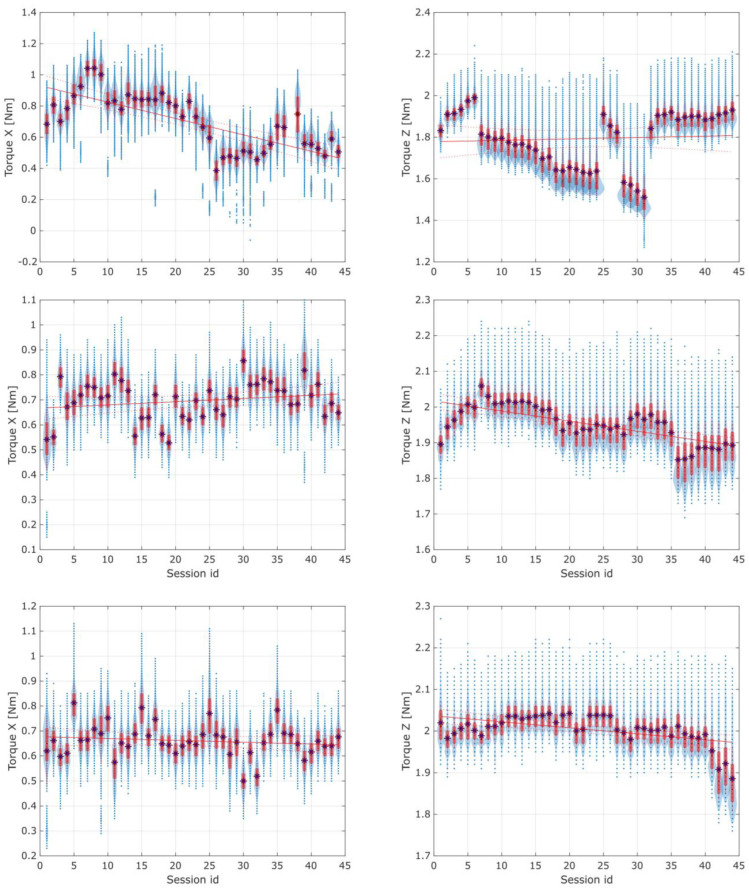
Torque distribution plot for 44 sessions of the three spur gears tested. From top to bottom, the successive horizontal pairs of plots reference the tested gears.

**Figure 18 materials-16-05554-f018:**
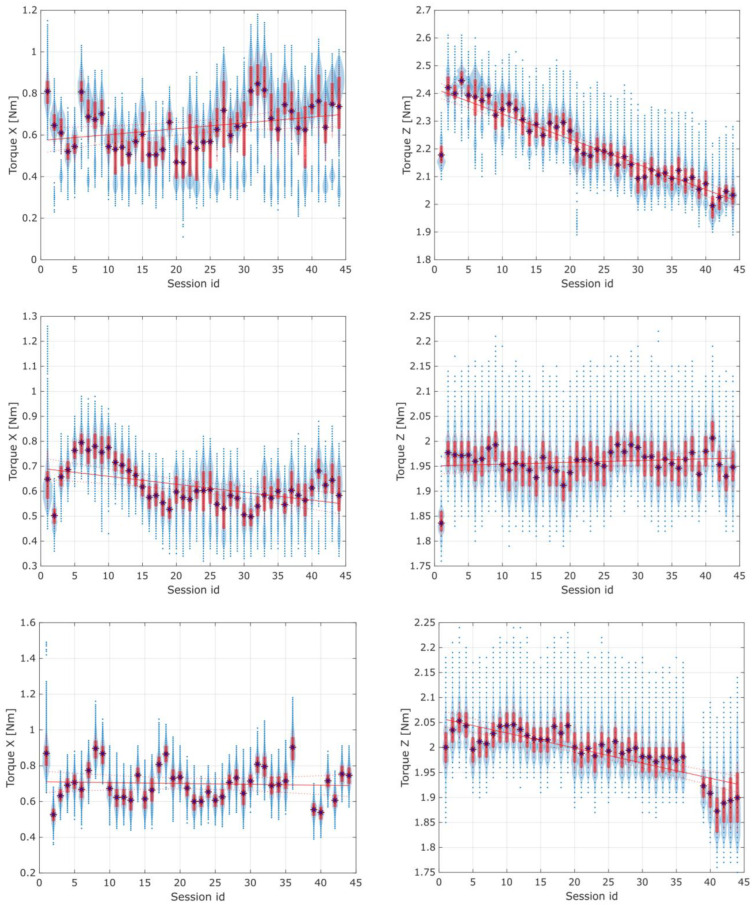
Torque distribution plot for 44 sessions of the three herringbone gears tested. From top to bottom, the successive horizontal pairs of plots reference the tested gears. Note: Sessions 36 and 37 for the last herringbone gear tested are missing due to temporary experimental setup failure—all recorded channels were zeroed out and retracted from the data frame.

**Figure 19 materials-16-05554-f019:**
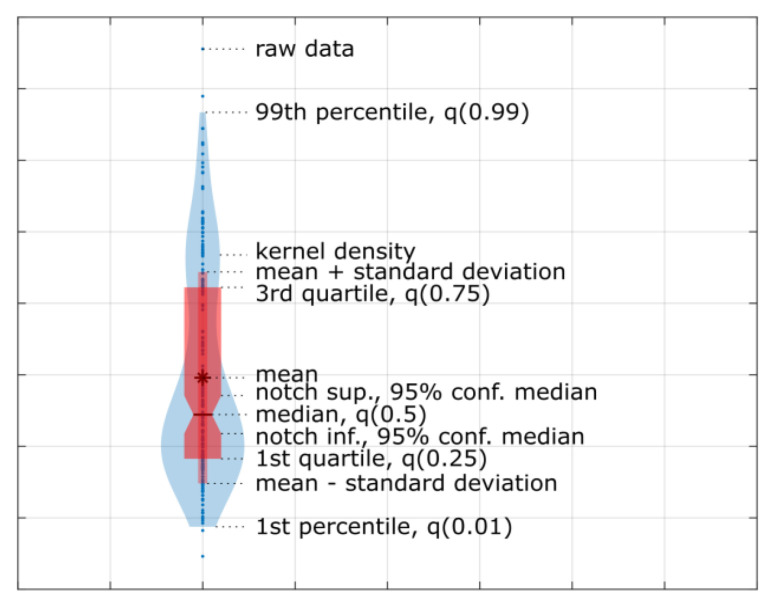
Torque distribution plot annotations.

**Figure 20 materials-16-05554-f020:**
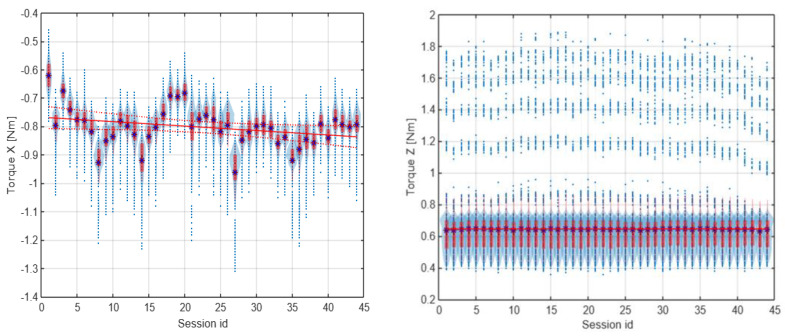
Torque profile for the idle position in the *X* and *Z* axes for the first spur gear tested.

**Figure 21 materials-16-05554-f021:**
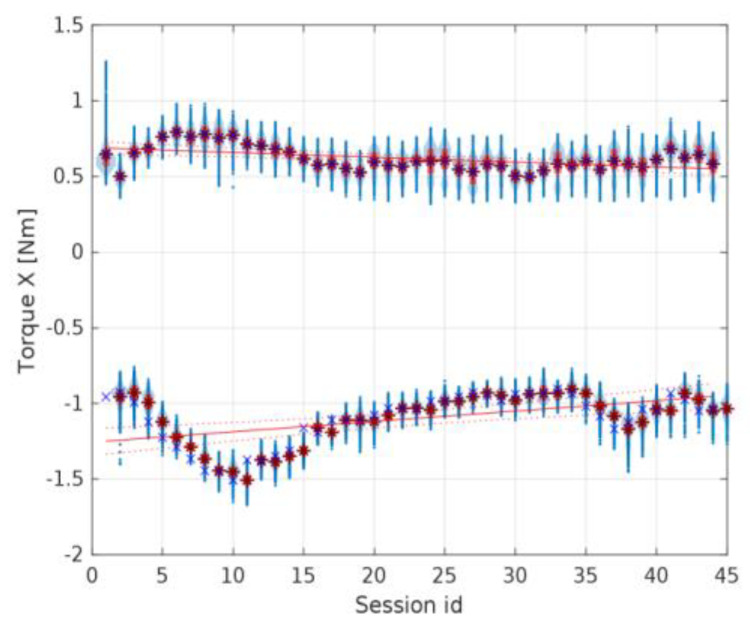
Torque profile for the work and idle position in the *X*-axis for the first herringbone gear tested with the measured values visualized at the 0.5 Nm level and around −1 Nm, respectively.

**Figure 22 materials-16-05554-f022:**
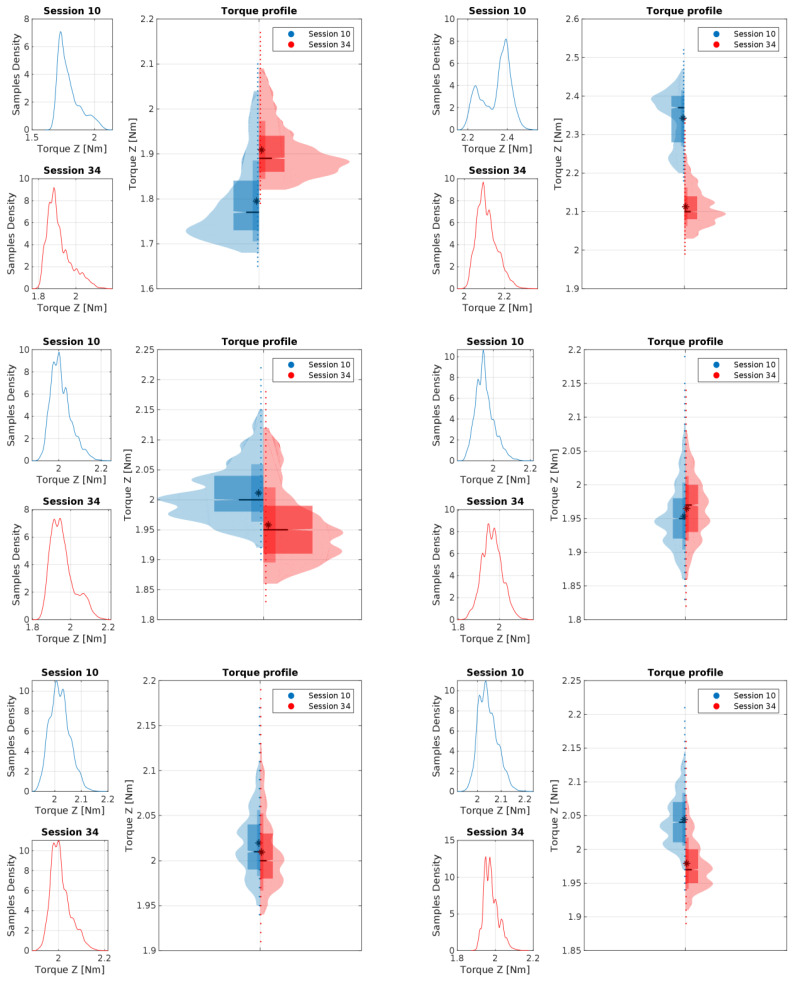
Z-axis torque profiles for the work position at the 0.5 Nm level and the idle position (at around −1 Nm), compared for sessions 10 and 34 for the three spur gears (left column) and the three herringbone gears (right column). The sign * (asterisk) refers to the mean value.

**Table 1 materials-16-05554-t001:** The values of measured geometrical parameters of the unexploited and exploited spur and herringbone gears. Designed dimensions are provided for reference.

Condition	Gear Type and No.	Measured Outer Circle Diameter [mm]	Measured Inner Circle Diameter [mm]	Measured Outer Circle Center Coordinates		Measured Inner Circle Center Coordinates		Calculated Outer and Inner Circle Centers Distance [mm]
				X-Axis [mm]	Y-Axis [mm]	X-Axis [mm]	Y-Axis [mm]	
Unexploited	Spur 1	53.24	19.86	32.49	30.30	32.50	30.30	0.02
	Spur 2	53.29	20.20	31.36	31.33	31.21	31.33	0.16
	Spur 3	53.11	19.87	29.62	29.53	29.68	29.55	0.06
Exploited	Spur 1	53.29	19.86	29.78	31.66	29.79	31.62	0.05
	Spur 2	53.40	19.94	27.17	28.17	27.16	28.22	0.04
	Spur 3	53.28	19.90	32.11	30.55	32.08	30.52	0.04
Unexploited	Herringbone 1	53.21	19.87	31.31	30.61	31.24	30.63	0.08
	Herringbone 2	53.12	19.94	35.03	30.76	35.10	30.70	0.09
	Herringbone 3	53.21	19.91	31.41	28.89	31.41	28.89	0.00
Exploited	Herringbone 1	53.23	19.88	32.51	29.89	32.57	29.90	0.06
	Herringbone 2	53.33	19.91	31.82	30.10	31.79	30.08	0.03
	Herringbone 3	53.31	19.86	31.72	31.02	31.73	30.99	0.03
Designed dimensions		54.00	20.00	-	-	-	-	0.00

**Table 2 materials-16-05554-t002:** Values of the remaining calculated and measured geometrical parameters of the unexploited and exploited spur and herringbone gears. Designed dimensions are provided for reference.

Condition	Gear Type and No.	Calculated Pitch Circle Diameter [mm]	Calculated Modulus [mm]	Calculated Circular Pitch [mm]	Measured Tooth Whole Depth [mm]	Measured Addendum [mm]	Measured Dedendum [mm]
Unexploited	Spur 1	48.60	1.97	6.11	4.57	2.31	2.27
	Spur 2	47.81	1.97	6.01	4.55	2.65	1.90
	Spur 3	48.50	1.97	6.09	4.62	2.36	2.26
Exploited	Spur 1	48.37	1.97	6.08	4.65	2.20	2.41
	Spur 2	48.73	1.98	6.12	4.68	2.31	2.31
	Spur 3	48.84	1.97	6.14	4.57	2.23	2.36
Unexploited	Herringbone 1	48.84	1.97	6.13	4.61	2.46	2.41
	Herringbone 2	48.67	1.97	6.11	4.62	2.34	2.34
	Herringbone 3	48.66	1.97	6.11	4.59	2.22	2.35
Exploited	Herringbone 1	48.87	1.97	6.14	4.56	2.18	2.38
	Herringbone 2	48.77	1.98	6.13	4.65	2.28	2.37
	Herringbone 3	48.76	1.97	6.12	4.64	2.28	2.37
Designed dimensions		50.00	2.00	6.283	4.40	2.00	2.40

## Data Availability

Data will be made available on a reasonable request.

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
