# Peer review of "Wear Analysis of 3D-Printed Spur and Herringbone Gears Used in Automated Retail Kiosks Based on Computer Vision and Statistical Methods"

_materials, 2023, doi:10.3390/ma16165554_

Round 1
Reviewer 1 Report
A potential symptom of the 3D-printed gearwheel technical condition (health) that is practical in the working environment was found. Several comments given to the authors, please find it below.
General comments:
1. The Reviewer does not see the novel in the present article. My examination revealed that several similar previous publications appear to appropriately address the issues you have brought up in the current submission. Please emphasize it more advance in the introduction section if there are any more truly something really new.
2. Please explain some effort to reducing wear in previous study, it would be from fabrication processes as explained by Farooq et al., materials selection as explained by Ammarullah et al., and geometric parameter as explained by Tauviqirrahman et al. Provide this information and supported with relevant reference as follows: https://doi.org/10.3390/ma16124458, https://doi.org/10.3390/biomedicines11030951, and https://doi.org/10.1038/s41598-023-30725-6
3. Because of grammatical faults and linguistic style, the authors must proofread the document.
Specific comments:
1. Line 53, please give a more explanation of manufacturing cost.
2. Line 81, the term use “polymer (plastic)”, please just use “polymer”.
3. Line 118-124, the part feel not unnecessary, recommended to delete it.
4. Line 125, as an important parameter in printer 3D configuration. The g code and slicing methods needs to justify.
5. Line 126, more detail of 3d printing parameter is needed.
6. Line 137, what is the reasons for choose ATMAT Signal Pro 3D? Please give a rationaltiation. It is the best product in their class? Or anything?
7. Line 143, please provide engineering drawing with the dimension.
-
Reviewer 2 Report
The authors investigated the wear properties of 3D printed gears. While the manuscript is generally well executed, there are several issues that should be addressed before further consideration for publication.
1. Suggest the authors to use ISO/ASTM terminology when describing the additive manufacturing process. There should also be details on the manufacturing process of the specimens as there are numerous AM techniques.
- Wickramasinghe et al. (2022), Flexural behavior of 3D printed bio-inspired interlocking suture structures, Materials Science in Additive Manufacturing 1 (2), 9
- Goh et al. (2020), Process–structure–properties in polymer additive manufacturing via material extrusion: A review, Critical Reviews in Solid State and Materials Sciences 45 (2), 113-133
2. What are the process parameters used to fabricate the gears? These have effect on the properties and should be clearly defined.
3. Any discussion on the deviation between the fabricated samples and the design? Will these affect the results discussed?
4. Should the results be presented using statistical analysis? With average etc? How many replicates are used?
5. Any discussion on the material properties? Do we expect similar results if the feedstock for fabrication used is different?
Reviewer 3 Report
This paper examines well the wear assessment conducted on prototype spur and herringbone gears manufactured using PET-G filament through additive manufacturing. The primary goal of this study is to determine the reliability of 3D-printed gears for long-term use in automated retail kiosks. To achieve this, two methods were employed: (1) visually inspecting the cross-sectional geometry of the gears encased in epoxy resin, which involves destructive testing to identify any changes in shape caused by operation, and (2) statistically characterizing the kinematic parameters and torques generated by the gear drives, aiming to find non-destructive indicators of wear. The results indicate that neither of the measurement techniques revealed significant damage that would prevent the tested gears from continued use after a simulated one-year period. The paper contains an abundance of data but lacks sufficient reasoning and explanations. While the paper is deemed acceptable, it does require a few corrections:
1. The authors have stated that epoxy resin does not undergo plastic deformation at high temperatures, but this claim needs to be substantiated with appropriate references or evidence.
2. Due to PET-G's glass transition temperature of 85°C, it becomes a matter of concern when cutting it, as the temperature can have an impact when it exceeds this threshold. The authors should specify the temperature and cutting instrument used to address this uncertainty.
3. Table 1 and 2 should include the designed dimensions of the gears, as well as the counts of errors observed after 3D printing.
4. Please review Figure 18, as both plots pertain solely to the Z-torque of the third gear.
5. It is strongly advised to merge the results and discussion sections for better coherence. The numerous figures contain extensive data, but there is a lack of proper explanations for the observed trends and results. The authors should combine these sections and provide comprehensive reasoning for each plot. For instance, in Figure 19, there are multiple terms that remain unaddressed and require discussion.
6. There is no need to present a research gap in the abstract (line 20-23)
7. The abstract must contain some significant results from the study and not just texts, reasons, affirmations, affirmations, statements, objectives, research gap, etc.
8. Keywords seem inappropriate, avoid short-forms.
Round 2
Reviewer 1 Report
Good job to the authors in the revised version, but several remarks still needs to addressed as follows:
1. Line 31, overall quality of the manuscript in introduction section was improved. But the reviewer does not see the part that discuss the significant novel in the present work. Please explain it properly.
2. Line 36, what it is mean with “27/7 accessibility”? it feels not scientific sound, recommended to revise it.
3. Line 60, since the authors introduced fused deposition modelling (FDM) type printer. Please explain several types of printers first before specifically direct in FDM. The strong reasons with advantages analysis needs to introduced.
4. Line 70, please state biomedical industry too.
5. Line 150-155, please state it in narrative form, not point-by-point as present.
6. Line 181-183 seems not strong since there are many products that have this specification in the market. Give more critical explanation please.
7. Since the present manuscript performs experimental testing of mechanical component, please provide further study performing computational simulation. This approach brings several advantages such as lower cost and faster results compared to experimental testing. Please provide this information along with relevant reference as follows: https://jurnaltribologi.mytribos.org/v33/JT-33-31-38.pdf, https://doi.org/10.3390/su142013413, and https://doi.org/10.3390/ma14247554
-
Reviewer 2 Report
NA
Reviewer 3 Report
The authors have incorporated the suggested changes from the reviewers and the paper looks good now for further recommendation.
